# Antiviral Activity of Essential Oils Against Avian Influenza Virus H7N3 In Vitro and In Ovo Models

**DOI:** 10.3390/v17111464

**Published:** 2025-10-31

**Authors:** Inkar Castellanos-Huerta, Jaime A. Ángel-Isaza, Lucio Bacab-Cab, Kevin Yam-Trujillo, Alejandro Aranda, Sindi Alejandra Velandia-Cruz, Loufrantz Mendez, Victor M. Petrone-Garcia, Guadalupe Ayora-Talavera, Álvaro José Uribe

**Affiliations:** 1Department of Poultry Science, University of Arkansas, Fayetteville, AR 72701, USA; 2Promitec S.A., Bucaramanga, Santander 680001, Colombia; innovacion@promitec.com.co (J.A.Á.-I.); virologia@promitec.com.co (S.A.V.-C.); fisicoquimica@promitec.com.co (L.M.); comercial@promitec.com.co (Á.J.U.); 3Centro de Investigaciones Regionales, Dr. Hideyo Noguchi, Universidad Autonoma de Yucatan (UADY), Merida 97000, Yucatan, Mexico; lucio.bacab@correo.uady.mx (L.B.-C.); jorge.yam@correo.uady.mx (K.Y.-T.); talavera@correo.uady.mx (G.A.-T.); 4Maestria en Salud y Produccion Animal Sustentable, Facultad de Ciencias Naturales, Universidad Autonoma de Queretaro, Av. de las Ciencias S/N, Juriquilla, Santiago de Queretaro 76230, Queretaro, Mexico; aaranda04@alumnos.uaq.mx; 5Centro de Enseñanza Agropecuaria (CEA), Facultad de Estudios Superiores Cuautitlan, Universidad Nacional Autonoma de Mexico, Cuautitlán 54714, Estado de Mexico, Mexico; petrone@unam.mx

**Keywords:** essential oils, avian influenza virus H7N3, antiviral activity, in vitro, in ovo

## Abstract

The poultry industry is continually seeking efficient, practical strategies to control infectious diseases. Among these new alternatives are essential oils (EOs), naturally occurring compounds with antimicrobial properties. Their effectiveness has been demonstrated in various studies that focus on their broad antiviral properties. The present experiment evaluated the antiviral efficacy of an EOs formulation against the H7N3 subtype of avian influenza virus (AIV) by directly mixing virus and EOs (virus/EOs mixture) through an in vitro model in cultured Madin-Darby canine kidney cells (MDCKs). The experiment used a focus reduction neutralization test (FRNT) to determine the 50% inhibitory concentration (IC_50_) by virus/EOs mixture, as well as the application of EOs 24 h before infection, through a viral inhibition test using a chicken embryo (CE) in ovo model. The results demonstrated the antiviral activity of the EOs formulation against the H7N3 in vitro model (IC_50_ values of 20.4 and 38.3 ppm and selective index (SI) values of 9.4 and 5.1) and in ovo model (decreasing hemagglutination titers to 1 HA unit, 10^5.28^ embryo infectious dose 50% (EID_50_) per mL, and viral loads to approximately 10^11.4^ copies/mL) when applied in CE, 24 h before viral infection, representing the lowest replication indicators recorded during the experiment. According to the results, the EOs formulation demonstrated antiviral activity against AIV H7N3 both as a virus/EOs mixture and through application in ovo 24 h before infection. Application 24 h before infection in CE showed a significant effect compared with the virus/EOs mixture, demonstrating an antiviral effect in the ovo infection model. This study demonstrates both the virucidal and antiviral capacity of the compounds in the EOs formulation against AIV H7N3 and their efficacy when applied 24 h before infection in the in ovo model.

## 1. Introduction

The poultry industry is actively seeking new, practical strategies to control pathogens, including natural options such as herbal compounds. These include essential oil (EO) compounds with antimicrobial properties [1]. EOs are concentrated, volatile aromatic molecules with a complex chemical composition, including terpenes, aldehydes, ketones, esters, and alcohols, derived from plants through various extraction methods [2].

In veterinary medicine their use is widespread, enhancing production conditions—particularly in poultry farming—taking advantage of their antimicrobial benefits, antioxidant activity, and antiviral properties [3,4]. Active combinations of compounds derived from EOs (e.g., carvacrol, thymol, and p-cymene) demonstrated antimicrobial synergistic effects, indicating a potential natural strategy for combating infectious diseases [5]. They show antiviral effects, which differ significantly due to their non-specific properties and different mechanisms of action. These effects include damaging viral envelopes, interfering with viral adsorption and penetration [6,7], either by inactivating viral particles through direct exposure or causing irreversible changes to their structure [8], and finally, inhibiting viral replication or entry into host cells [9]. Both effects make EOs a promising approach for combating viral infections [10]. Examples of the ability of EOs to reduce the viral loads of significant viral diseases include influenza virus (IV) (H1N1), human herpes viruses (HSV), human immunodeficiency virus (HIV), coxsackie virus, bovine viral diarrhea virus, respiratory syncytial virus, yellow fever virus, adenovirus-3, bovine herpes virus 2, Zika virus, and avian influenza virus (AIV), where EOs of *Melissa officinalis* and *Fortunella margarita* have shown antiviral activity against subtypes H9N2 and H5N1, respectively [11].

Previous in vitro studies describe the antiviral capacity of EOs against AIV, related to affecting viral membranes [6,12], inhibition of the expression of viral NP and NS1 proteins [13], inhibition of membrane fusion and interaction of hemagglutinin with cell receptors [14], and demonstrate that the synergistic effects of EOs usually enhance their antiviral activity effect [7,15,16]. Therefore, the use of EO formulations are of interest to achieve antiviral effects at different stages of the viral cycle.

The EOs of *Lippia* spp., *Rosmarinus officinalis,* and *Eucalyptus globulus* possess remarkable antiviral properties, each with unique mechanisms and effects [11,17,18,19]. This antiviral activity is attributed to their complex chemical composition, which favors their antiviral effects by acting at various stages of the viral life cycle, directly affecting enveloped viral membranes and interfering with viral uptake and penetration, as is the case with *Lippia spp*., inhibiting viral RNA or DNA replication and interfering with viral uptake and penetration [6,11], and in the case of *R. officinalis* and *E. globulus* by disrupting viral envelopes and inhibiting the replication of respiratory viruses [6,20]. Therefore, an EOs formulation based on this combination may be of interest for evaluating its antiviral activity against AIV.

Among the various subtypes of AIV, H7N3 is one of the most important in poultry and is found in several countries, including Mexico [21]. Since the detection of the highly pathogenic avian influenza virus (HPAIV) in Mexico in 2012, its continued presence has been documented [22,23], resulting in economic impacts in the poultry sector and potential zoonotic risk. Current strategies for controlling AIV include biosecurity measures and vaccination with low-pathogenic avian influenza virus (LPAIV) strains or strains produced by biotechnological techniques [24]. However, ongoing antigenic evasion due to vaccination and the potential decrease in vaccine efficacy under specific conditions may result in additional outbreaks of AIV [25].

Despite the extensive implementation of these strategies, this viral subtype remains a potential threat to society and the poultry industry [22]. Therefore, AIV requires new and practical antiviral management strategies that are commercially available, have a broad antiviral spectrum not dependent on specific viral serotypes or mutations, and fit common practices in the poultry industry, such as using EOs (sprinkling or adding to drinking water). This requirement arises from the significant impact of AIV on the global poultry industry and its substantial economic consequences [21].

This scientific study examines the antiviral effect of a commercial EOs formulation against the AIV subtype H7N3, using both an in vitro culture of Madin-Darby canine kidney (MDCK) cells and a more robust, complex viral replication model, chicken embryos (CEs). The in ovo system, which involves CE, provides an effective environment for evaluating safety levels and the antiviral capacity of EOs within a highly efficient viral replication system. The objective was to determine the ability of the EOs formulation to reduce mortality associated with viral inoculation, decrease viral replication, and evaluate the antiviral effects of both simultaneous and residual EO applications in the in ovo model.

## 2. Materials and Methods

### 2.1. Essential Oils Formulation and Characterization

The EOs formulation consisted of a commercial mixture of EOs (Kontra^®^; Promitec, Santander, Colombia), approved by the Colombian Agricultural Institute (ICA) in 2019 (patent CO2020013930A1). This product is a topical spray nanoformulation composed of EOs from *Lippia* spp., *Rosmarinus officinalis,* and *Eucalyptus globulus*, whose phytochemical constituents—such as terpenes and phenolic compounds—have been widely reported for their antiviral activity [6,11]. The EOs were obtained by microwave-assisted hydrodistillation. Leaves and stems were previously air-dried in the dark, ground, and homogenized before processing. A 100 g portion of each plant material was distilled with 500 mL of water for 3.5 h using a Clevenger apparatus at atmospheric pressure, maintaining the temperature above 180 °C. After condensation, phase separation occurred due to differences in volatility and immiscibility, yielding the EOs [26].

Gas chromatography–mass spectrometry (GC–MS) was employed to analyze the chemical profile of each oil. The analysis was performed using electron ionization (70 eV) in full-scan mode using a nonpolar DB-5MS (5%-Ph-PDMS) column (J&W Scientific, Folsom, 60 m × 0.25 mm × 0.25 µm). Tentative compound identification was performed through spectral comparison and confirmed using reference standards, including a certified C6–C25 hydrocarbon mixture (AccuStandard, New Haven, CT, USA), fragmentation pattern matching against the Adams, Wiley, and NIST databases, and available reference substances. The identified compounds for each EOs are listed in Table 1. The obtained EOs were subsequently incorporated into an oil-in-water (o/w) nanoemulsion and stabilized with natural emulsifiers that maintained the hydrophilic–lipophilic balance (HLB), ensuring long-term droplet stability and homogeneity [27].

For this study, the EOs formulation was emulsified in distilled water to obtain a stock solution at a final concentration of 6000 ppm, which is four times the dose recommended by the supplier, to assess safety levels and potential effects on CEs during the experiment. It was then diluted to prepare the concentrations tested in the experiments.

### 2.2. Viruses and Cells

The reverse genetic reassortment AIV strain A/chicken/Guanajuato/CPA-06664-18-VS/2018 LPAIV (CPA) was used to assess the antiviral effect of EOs in vitro and in ovo assays. Additionally, the recombinant AIV strain A/chicken/Queretaro/01/2018 (H7N3), identified as WT (for experimental identification purposes only), was used for in vitro testing as a biological reference and internal control for comparison of pathogenicity/replication against the CPA vaccine virus. Reverse genetics viruses were generated as previously described Pérez (2024) [28]. CPA and WT samples were cultured in MDCKs (ATCC CCL-34, American Type Culture Collection, Manassas, VA, USA), propagated in Dulbecco’s modified Eagle media (DMEM; Gibco, Thermo Fisher Scientific, Waltham, MA, USA) supplemented with 10% fetal bovine serum (FBS; Gibco, Thermo Fisher Scientific, Waltham, MA, USA), titrated by plaque assay, aliquoted, and stored at −80 °C until further use.

### 2.3. In Vitro Cytotoxicity Assay of the EOs Formulation Using MDCKs

The 50% cytotoxic concentration (CC_50_) assay was performed on MDCKs using the 3-[4,5-dimethylthiazol-2-yl]-2,5-diphenyltetrazolium bromide (MTT) cell viability assay, as previously described [29]. Briefly, cells were seeded in sterile 96-well polystyrene plates at a density of 1 × 10^4^ cells per well in DMEM with 10% FBS and left to incubate for 24 h at 37 °C in a 5% CO_2_ atmosphere until confluency. Subsequently, cells were washed twice with phosphate-buffered saline (PBS) at pH 7.2 and incubated with four concentrations (50, 100, 300, and 500 ppm) within the range reported for cytopathic effects of EOs with similar terpene profiles to determine the most suitable and safe concentration for cells during this experiment. Consequently, a similar range was established for this initial trial [30]. Each concentration was tested in quadruplicate.

The cells were incubated with the EOs formulation for 72 h. After incubation, cell viability was assessed using the MTT assay. A total of 10 µL of MTT (Sigma-Aldrich, St. Louis, MO, USA) reagent was added to the cell plates without removing the supernatant containing different concentrations of the EOs formulation. Plates were incubated for 4 h at 37 °C. Soon after, 100 µL of solubilization buffer was added per well, and the plates were incubated overnight at 37 °C in a 5% CO_2_ atmosphere. After incubation, plates were checked for complete solubilization of the purple formazan crystals, and absorbance was measured using a microplate enzyme-linked immunosorbent assay (ELISA) reader. The wavelength used to measure the absorbance of the formazan product was between 550 and 600 nm, as determined by the filters available for the ELISA reader, with a reference wavelength of greater than 650 nm. To determine the CC_50_, cell viability was determined by comparing the optical density (OD) of the cells in the presence of the different concentrations of the EOs formulation and the cell control, which was taken as 100% of viable cells [% viability = (OD treated cells/OD cell control) × 100]. The concentration CC_50_ was determined by plotting the concentration of the EOs formulation against the viability percentage and obtaining it by regression analysis. Untreated cells and cells treated with Triton X-100 (0.1%, Sigma-Aldrich, St. Louis, MO, USA) were run in parallel as negative and positive controls, respectively. Each concentration was analyzed in three independent assays, each performed in triplicate.

### 2.4. Antiviral Activity of the EOs Formulation by Focus Reduction Neutralization Test in MDCKs

The antiviral activity of the EOs formulation against AIV H7N3 was assessed by quantifying the reduction in infectious foci using a modified focus reduction neutralization test (FRNT) [31]. The viral inoculum was standardized to 100 infectious foci per well as the initial challenge dose [32].

MDCKs were seeded in a sterile 96-well polystyrene plate at a cell density of 1 × 10^4^ cells per well in DMEM containing 10% FBS and incubated for 24 h at 37 °C in a 5% CO_2_ atmosphere until confluence was achieved. Then, the cells were washed twice with PBS and incubated with the virus/EOs mixture. Each virus (CPA and WT) was tested separately. Based on the cytotoxicity test results, a maximum safe level of 100 ppm was established for use in MDCKs. It was suggested to use standard dilutions to evaluate its potential antiviral effect without causing cellular damage from EOs application. The final concentrations of the EOs formulation were 3.12, 6.25, 12.5, 25, 50, and 100 ppm. All mixtures of virus/EOs were pre-incubated for 1 h at 37 °C before being added to the cells.

Following inoculation, plates were incubated for 48 h in DMEM supplemented with 2.5% carboxymethylcellulose and TPCK-treated trypsin (2 µg/mL; Sigma-Aldrich, St. Louis, MO, USA). After incubation, the cells were fixed and then incubated for 1 h at room temperature with a primary anti-NP influenza A virus antibody (Invitrogen, Thermo Fisher Scientific, Waltham, MA, USA) at a concentration of 1 µg/mL in PBS/Tween. They were then washed four times with a PBS/Tween solution and incubated for 1 h at room temperature with an HRP-conjugated anti-mouse IgG secondary antibody (Jackson ImmunoResearch, West Grove, PA, USA) at a dilution of 1:2000. After incubation, the cells were rewashed four times with PBS/Tween. TrueBlue™ peroxidase substrate (Sera Care, Milford, MA, USA) was then added to develop a color reaction according to the manufacturer’s instructions.

To determine the IC_50_, the plates were read on a Bio Reader 7000-E (Biochrom, Cambridge, UK), and graphs were generated using GraphPad Prism v9 (GraphPad Software, San Diego, CA, USA). The data were modeled using a nonlinear regression approach via the function [log(inhibitor) vs. response—variable slope (four parameters)]. All experiments were performed in three independent trials, each with triplicate measurements. Finally, the selective index (SI) values (SI = CC_50_/IC_50_) were calculated.

### 2.5. Assessment of the Toxicity of the EOs Formulation in Chicken Embryos

The toxicity test was conducted to determine the highest non-toxic dose, defined as the concentration of EOs resulting in >90% survival of CEs. For this assay, the CEs from commercial broilers (Ross 308) were obtained from a local hatchery in Yucatan, Mexico, and incubated under standard conditions until use in the experiment. Before inoculation, CEs were candled and subsequently randomly assigned to the experimental groups. The toxicity test consisted of applying the EOs formulation via the allantoid cavity in a volume of 0.1 mL. A total of eighty 9-day-old CEs were used and randomly divided into eight experimental groups (*n* = 10 per group). The groups were as follows: group A (4800 ppm), group B (2400 ppm), group C (1500 ppm), group D (1200 ppm), group E (600 ppm), group F (350 ppm), group NTC (0.1 mL of PBS-inoculated), and NC group (non-intervention group). All groups were monitored every 24 h to determine mortality. Incubation continued until mortality was observed or until 96 h post-application, whichever occurred first. The toxicity test was conducted in accordance with the Mexican Official Standard NOM-062-ZOO-1999, which establishes the technical specifications for the production, care, and use of laboratory animals and management of CEs in the laboratory. According to this regulation, the use of this in ovo model does not require the approval of a bioethics committee in Mexico [33].

### 2.6. Viral Inhibition Assay of the EOs Formulation Against AIV H7N3 in Chicken Embryos

The viral inhibition test was performed using the highest non-toxic dose of the EOs formulation, based on the >90% survival of CEs observed in the experiment (1500 ppm). Two experimental approaches were evaluated: (i) treatment applied to CE 24 h after AIV inoculation; (ii) direct exposition of the virus to the EOs formulation before inoculation (virus/EOs mixture).

The viral challenge consisted of individually inoculating 0.1 mL of CPA virus at an adjusted titer of 10^7^ embryo infectious dose 50% (EID_50_)/0.1 mL, corresponding to 10^12.02^ mean viral load copies per 1 mL. Four groups of ten 9-day-old commercial broiler CEs (Ross 308) were included in the experiment. CEs were candled and randomly assigned to the following treatments: group T1 (EOs formulation applied 24 h before the viral challenge), group T2 (virus/EOs mixture at a 1:1 ratio), group H7v (positive control, inoculated via allantoic cavity with 0.1 mL of CPA at 10^7^ EID_50_/0.1 mL), and group NTC2 (negative control, inoculated with 0.1 mL of PBS). All groups were monitored every 24 h to record mortality.

Incubation continued until either mortality was observed or 72 h had elapsed since inoculation, whichever came first. EC mortality within the first 24 h post-inoculation was excluded from this analysis.

Mortality was evaluated by a hemagglutination assay [32] to verify viral replication, and further analyzed by quantitative real-time PCR (qPCR) to determine the log_10_ mean viral copies per mL [34] and EID_50_/mL [32] for the pooled mortality for each treatment. All samples were stored at −80 °C until further use. As previously described, the viral inhibition test was conducted in compliance with the Mexican Official Standard NOM-062-ZOO-1999 [33].

### 2.7. Statistical Analysis

Nonlinear regression analysis of the four-parameter logistic model was applied to calculate the CC_50_ and IC_50_ values using GraphPad Prism v9 (GraphPad Software, San Diego, CA, USA). Kaplan–Meier survival analysis was performed to estimate the probability of survival over time in each group in the viral inhibition test in CE. The log-rank test was used to evaluate statistical comparisons between groups using JMP software version 17.0 (SAS Institute, Cary, NC, USA). qPCR for viral loads in CE values was analyzed using GraphPad Prism v10 (GraphPad Software, San Diego, CA, USA). Comparisons between treatments and controls were performed using one-way ANOVA with Tukey’s post hoc test; results are presented as mean ± SEM, and samples not sharing the same letter were considered significantly different (*p* < 0.05).

## 3. Results

### 3.1. In Vitro Cytotoxicity Assay of the EOs Formulation Using MDCKs

The cytotoxicity assay revealed that concentrations of 300 and 500 ppm of the EOs formulation induced complete cell death in all treated MDCKs. In contrast, no cytotoxicity was observed at 100 ppm, with a 100% cell viability—the CC_50_ of the formulation was determined to be 196 ppm (Figure 1A).

Subsequently, the antiviral effects were evaluated using concentrations below the CC_50_. The following six concentrations were tested: 3.12, 6.25, 12.5, 25, 50, and 100 ppm. The EOs formulation exhibited antiviral activity in a dose-dependent manner. The IC_50_ values were 20.4 ± 1.1 ppm for CPA and 38.3 ± 1.2 ppm for WT, corresponding to SI values of 9.6 and 5.1, respectively (Figure 1B). At the highest concentration tested (100 ppm), the viral replication of all strains was inhibited by over 80% in MDCKs (Figure 2).

### 3.2. Assessment of the Toxicity of the EOs Formulation in Chicken Embryos

The EOs formulation exhibited low toxicity in the CE model. The highest concentration with no mortality via the allantoic cavity was 1500 ppm. At the highest doses, 2400 and 4800 ppm, toxicity was observed within 24–48 h, resulting in 40% mortality, which remained unchanged until 96 h. None of the tested concentrations caused mortality above 50% (Figure 3). Lower concentrations of 1200, 600, and 350 ppm caused only 10% mortality, comparable to that observed in the negative toxicity control (NTC; embryos inoculated with PBS). No mortality occurred in the negative control (NC) group.

### 3.3. Viral Inhibition Assay of the EOs Formulation Against AIV H7N3 in Chicken Embryos

The viral inhibition test with the CPA strain revealed differential effects of the EOs formulation on CEs over time. At 24 h post-challenge, cumulative mortality was 20% in groups T1 and T2 and 10% in group NTC2. By 48 h, mortality in T1 remained at 20%, whereas mortality in T2 increased to 40%, compared with 80% in the untreated control (H7v). At 72 h, cumulative mortality reached 80% in T1 and 60% in T2, while H7v showed 100% mortality. No mortality was recorded in group NTC2 at 48 or 72 h (Figure 4A). Hemagglutination assays performed at 24 h post-challenge showed no detectable hemagglutination in T1, T2, or NTC2. At 48 h, T2 exhibited an HA titer of five units, while H7v reached a titer of eight units. By 72 h, T1 had a HA titer of 1 HA unit, T2 maintained titers of 5 HA units, and H7v remained a titer of eight units (Figure 4B). Consistently, the mean EID_50_ titers at 48 and 72 h were markedly reduced in T1 (10^5.28^ EID_50_/mL) and T2 (10^5.57^ EID_50_/mL) compared to H7v (10^8.76^ EID_50_/mL).

Viral loads quantified by qPCR (log_10_ copies/mL) in CEs reflect these trends. At 48 h, H7v exhibited the highest replication (~10^12.43^), with a significantly lower viral load in T2 (~10^11.91^, *p* < 0.05) corresponding to a ~0.52 log_10_ (~70%) decrease in viral genome copies. By 72 h, viral loads decreased across treatments, with T1 exhibiting the lowest trend (~10^11.4^). Notably, T1 at 72 h showed a ~1.03 log_10_ reduction in viral load (~91%) compared with peak H7v replication (~10^12.43^ copies/mL at 48 h), which is biologically meaningful. No significant difference was observed among T1 (~10^11.4^), H7v (~10^11.61^), and T2 (~10^11.8^; *p* > 0.05).

These results demonstrated that H7v maintains the highest replication rate during the first 48 h, T1 effectively reduces viral replication by 72 h, and T2 exhibits an intermediate response, with lower initial viral loads at 48 h but more stability thereafter (Figure 5 and Table 2). This suggests a biologically relevant inhibition of viral replication within the CE environment.

**Figure 3 viruses-17-01464-f003:**
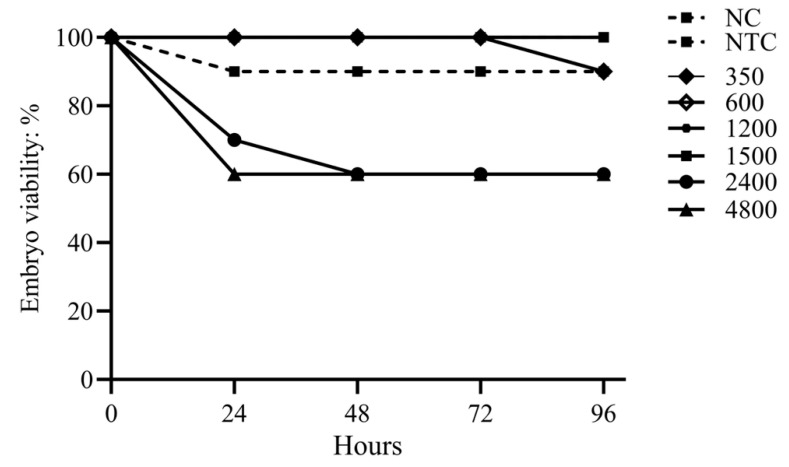
Toxicity of the EOs formulation in chicken embryo (in ovo). CE treatment (300–4800 ppm) and viability were monitored for up to 96 h. Non-inoculated and PBS-inoculated embryos served as controls (NC and NTC, respectively). Higher concentrations (4800 ppm and 2400 ppm) resulted in higher cumulative mortality within the first 48 h. Controls and treatments with concentrations ≤ 1500 ppm showed no significant mortality. Points indicate cumulative mortality rates per 24 h interval, and error bars represent the 95% confidence interval. Toxicity was defined as any group with a viability rate of 90% or less at the end of the trial.

**Figure 4 viruses-17-01464-f004:**
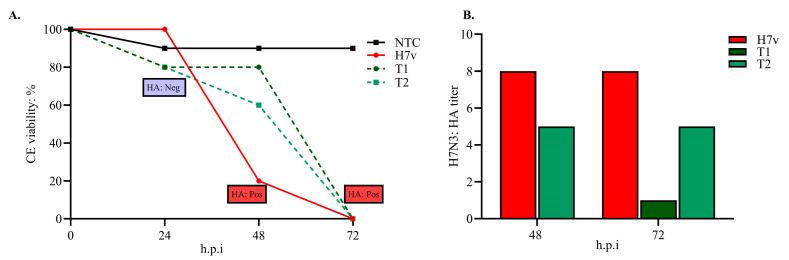
Viral inhibition activity of the EOs formulation in the chicken embryos (in ovo). CEs were treated with the EOs formulation either 24 h before infection (T1) or at the time of infection (T2) with the H7N3 CPA strain. Non-treated infected embryos (H7v) served as the viral control, and non-infected non-treated embryos (NTC) as the negative control. (**A**) Cumulative CE viability was monitored for 72 h after infection, and the presence of hemagglutination (HA) was assessed every 24 h in recorded mortality. The presence of HA positive at the time of death is marked in colored boxes. HA: Pos indicates the detection of hemagglutinating virus in allantoic fluid at the moment of embryo mortality. Each embryo was tested promptly after death was observed. (**B**) Hemagglutinin titers (H7N3: HA titer) in CEs exhibited in the T2 group a titer of 5 HA units, whereas the H7v group exhibited a titer of 8 HA units at 48 h. At 72 h post-infection, the T1 group demonstrated a titer of 1 HA unit, while the T2 and H7v groups retained titers of 5 and 8 HA units, respectively.

**Figure 5 viruses-17-01464-f005:**
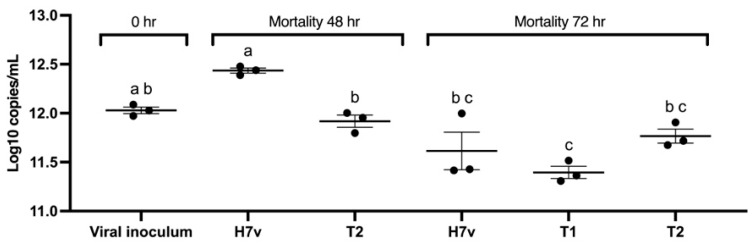
Viral loads in chicken embryos evaluated by qPCR (log_10_ copies/mL) at 48 and 72 h after infection. Each point represents an individual replicate (n = 3 per sample). The black horizontal line indicates the mean, and the vertical bars show the standard error of the mean (SEM). Statistical differences among groups were evaluated using one-way ANOVA followed by Tukey’s post hoc test. Groups that do not share letters are considered significantly different (*p* < 0.05). The viral inoculum (~10^12.03^ copies/1 mL) is shown for reference.

**Table 1 viruses-17-01464-t001:** Principal constituents of the essential oils formulation present at levels above 1%.

Compound	RT (min)	Amount (%)
1,8 cineole	20.7	7.5
Camphor	25.5	4.1
Geranial	29.8	2.8
β-pinene	18.2	2.2
Thymol	30.5	2.9
Carvacrol	30.9	1.7
α-pinene	16.3	1.9
Caryophyllene oxide	40.9	1.5
p-cimene	20.3	1.8

Identified volatile compounds from the essential oils showing relative proportions above 1%. Retention time (RT) is expressed in minutes, and compound amounts are presented as a percentage of the total identified components.

**Table 2 viruses-17-01464-t002:** Mean viral loads (log_10_ copies/mL) ± SEM in CE at 48 and 72 h before inoculation.

Treatment	Mean (log_10_ Copies/mL) ± SEM
Viral inoculum (0 h)	12.03 ± 0.03 a,b
H7v (48 h)	12.43 ± 0.02 a
T2 (48 h)	11.91 ± 0.05 b
H7v (72 h)	11.61 ± 0.16 b,c
T1 (72 h)	11.39 ± 0.054 c
T2 (72 h)	11.77 ± 0.061 b,c

Statistical differences were assessed using one-way ANOVA and Tukey’s post hoc test; treatments with different letters are significantly different (*p* < 0.05).

## 4. Discussion

Several antiviral approaches have been proposed, such as targeting viral polymerase proteins, ion channels, host receptors, or viral enzymes [11]; however, these have not been widely developed or adapted for testing in the poultry industry or in ovo model evaluation. EOs are promising and accessible alternatives for veterinary medicine [1] because they exhibit broad chemical variability and an antigen-independent mode of action [1].

These advantages suggest their potential use against AIV subtypes, particularly H7N3, which is known for frequent antigenic shifts [35]. Its antiviral effect has been tested in an in ovo model in the case of viral infections in poultry, particularly in relation to AIV [36,37,38,39,40,41], directly impacting viral replication by significantly reducing it. However, these experiments test the direct contact of the EOs and viral particles without exploring their antiviral effects before viral infection, which would be more relevant for preventing infections in field conditions; considering the ability of EOs to reduce viral entry into host cells, induce in ovo interferon responses via Toll-like receptor (TLR) ligands, and in some cases, be more effective than antivirals [40,42,43].

In the present work, the formulation of EOs (a mixture of *Lippia* spp., *R. officinalis*, and *E. globulus*) showed an antiviral activity against AIV H7N3 subtype by reducing viral infection in vitro by 50% in both HPAIV and LPAIV H7N3 strains at concentrations as low as 38 and 20 ppm, respectively; and resulting in IC_50_ values of 38.3 ppm (HPAIV) and 20.4 ppm (LPAIV), with SI > 4 [11]. The LPAIV strain showed superior IC_50_ and SI values, attributed to differences in replication dynamics between LPAIV and HPAIV strains [44].

In the in ovo model, the EOs formulation delayed CE mortality and reduced viral replication, as indicated by hemagglutination titers, EID_50_ titers, and qPCR-measured media viral load. At 48 h post-challenge, mortality in the treated groups ranged from 20 to 40%, compared with 80% in the viral control. By 72 h, cumulative mortality reached 100% in all groups. However, the antiviral effects of EOs were confirmed by hemagglutination titers, EID_50_, and qPCR-measured viral loads, with T1 showing viral load of ~10^11.4^ copies/mL and 10^5.28^ EID_50_/mL, followed by T2 with ~10^11.91^ copies/mL and 10^5.57^ EID_50_/mL, in comparison to the virus control with ~10^12.43^ copies/mL and 10^8.76^ EID_50_/mL. Although T1 and T2 showed slightly higher early mortality (20% at 24 h) compared with NTC2 (10%), this difference is likely related to early viral replication dynamics or inoculation stress rather than EO toxicity, as the overall embryo viability and development remained normal throughout the experiment.

These findings confirm that the formulation exerted an antiviral effect in ovo, decreasing both mortality and viral replication compared to the virus control at 48 and 72 h of the experiment respectably. Although embryonic mortality was not prevented, AIV H7N3 replication capacity was reduced by the antiviral effect of the EOs formulation, since both groups T1 and T2 showed lower viral replication than the H7v viral control during the experiment. These findings demonstrate that the EOs formulation exhibited both prophylactic and virucidal effects, with T1 (pre-treatment) showing the strongest inhibition of viral replication, while T2 consistently maintained lower viral titers than the control. The mechanism of action of EOs against AIV depends directly on the timing of their administration, with possible direct viral inhibition linked to changes in the components of AIV, including membranes and proteins such as HA and NA that connect with receptors. They may also affect replication by disrupting the release of genetic material within compartments, such as lysosomes, in the infected cells [11]. Therefore, their use before infection can broaden our understanding of how EO components work.

During the experiment, the T1 group, in which the EOs were applied before viral infection, showed better performance than the T2 group, in which the virus was pre-incubated with EOs. This is supported by viral loads of ~91% and ~70% in groups T1 and T2, respectively, relative to the control viral loads and those seen in the viral control group. These results are consistent with earlier reports, where EOs exhibit more substantial effects when used in early stages of replication, disrupting initial infection processes, reducing viral adsorption, damaging structural proteins, and preventing viral uncoating [11]. The reduction in viral replication in T1 and T2 confirms an antiviral effect, potentially through interactions between EOs and viral structures or alterations in host cell surfaces.

The antiviral activities of the components in the EOs formulation are consistent with previous reports, where EOs of *Eucalyptus globulus*, containing 1,8-cineole, limonene, α-pinene, and p-cymene, demonstrate antiviral effects against influenza-like infections both in vitro and in vivo [45,46]. In addition, *Rosmarinus officinalis* and *Lippia* spp. have shown antiviral activity against herpes simplex virus, bovine viral diarrhea virus, yellow fever virus, and dengue virus, with reported IC_50_ values ranging from 4 to 99 ppm [11]. Similarly to AIV, these viruses possess a lipidic membrane sensitive to lipophilic substances such as terpenes and phenolic constituents, which are present in EOs, supporting the comparable IC_50_ (20.4–38.3 ppm) range observed in this experiment.

The antiviral activity of EOs against avian viruses showed promising results when evaluated in an in ovo model, including Newcastle disease virus (NDV), infectious bronchitis virus (IBV), infectious bursal disease virus (IBDV), and AIV H5N1 [36,37,38,39,40,41]. However, the present study demonstrates that either inoculating AIV H7N3 with the EOs or applying EOs 24 h before infection in CEs leads to a reduction in viral replication, even at higher infective doses (10^8^ EID_50_/mL), compared to what was previously reported in AIV subtype H5N1 trials (10^3^ EID_50_/mL) [41].

The reduction in viral replication together with mortality compared to the viral control, observed in the group administered EOs prophylactically before infection, signifies a statistically significant enhancement; however, in the poultry industry, preventive treatment is anticipated to yield greater efficacy. The reduction in viral load and hemagglutination titers indicates that the formulation exhibits antiviral properties, which may be optimized in subsequent research.

## 5. Conclusions

In summary, this work demonstrated that an EOs formulation based on EOs from *Lippia* spp., *R. officinalis*, and *E. globulus* exerts antiviral effects against AIV H7N3 in vitro and in ovo. The antiviral effect is more significant when applied 24 h before infection, highlighting the potential of EOs as a complementary control strategy in AIV infections.

Despite the results obtained on mortality and viral replication in the in ovo model, these results should be considered preliminary. This work allows us to identify the antiviral potential of EOs; however, it does not fully represent the immune and metabolic response conditions of the in vivo model. Therefore, ethically appropriate in vivo experiments are required to establish safety, dose range, and practical application under poultry production conditions.

## Figures and Tables

**Figure 1 viruses-17-01464-f001:**
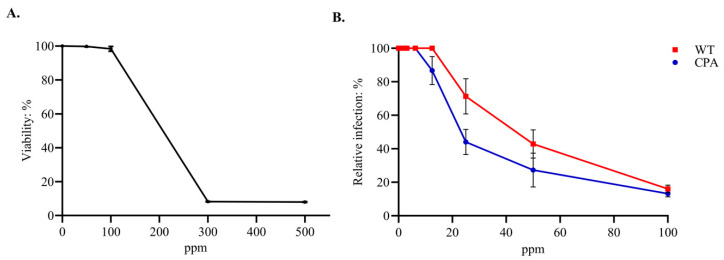
Cytotoxicity and viral inhibition activity of the EOs formulation in MDCKs. (**A**) Cytotoxicity of the EOs formulation: cells were incubated with 50, 100, 300, and 500 ppm of the EOs formulation for 72 h, and viability was measured using the MTT assay. (**B**) Viral inhibition activity of the EOs formulation against H7N3 AIV strains (CPA and WT): cells were treated with non-cytotoxic concentrations of the EOs formulation (3.12–100 ppm), and relative infection was determined by quantifying infectious foci using a modified FRNT. Data represent the mean ± SD of three independent experiments.

**Figure 2 viruses-17-01464-f002:**
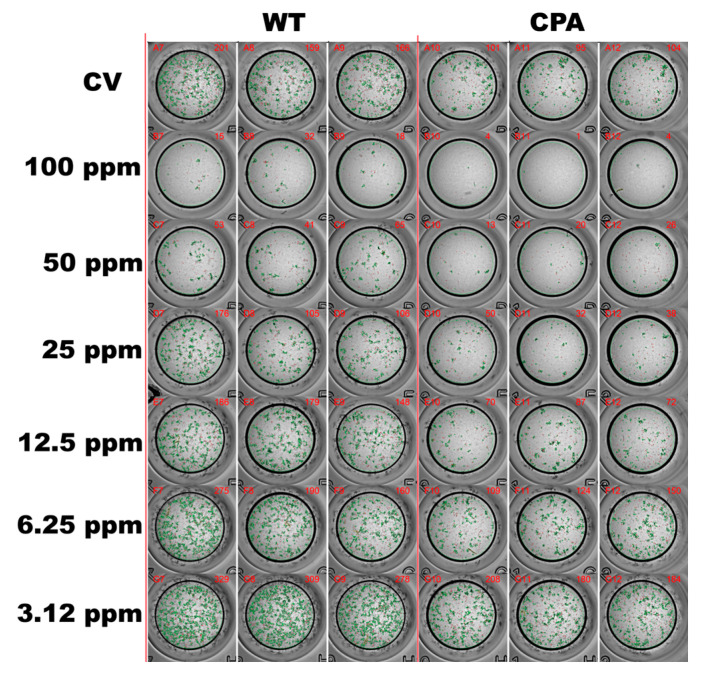
Assessment of the viral inhibition activity of the EOs formulation in MDCKs. The quantification of infectious foci was performed using a modified FRNT in MDCKs infected with AIV H7N3 virus strains WT and CPA using the EOs formulation at concentrations of 3.12, 6.25, 12.5, 25, 50, and 100 ppm. The control group (viral replication control) was designated CV.

## Data Availability

The data presented in this study are available on request from the corresponding author (the data are not publicly available due to privacy or ethical restrictions).

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
