# Peer review of "Antiviral Activity of Essential Oils Against Avian Influenza Virus H7N3 In Vitro and In Ovo Models"

_viruses, 2025, doi:10.3390/v17111464_

Round 1
Reviewer 1 Report
Comments and Suggestions for Authors
Adding a chromatographic profile of the essential oil composition or specifying the main components and their concentration is extremely important for reproducibility of the results.
Correct a typo in the caption to Figure 5: "48 and 72 hours before infection" → it should be "after infection".
To clarify during the discussion whether the observed reduction in mortality (80% in T1 versus 100% in control) is sufficient for practical application in poultry farming.

Author Response
Comment 1: Adding a chromatographic profile of the essential oil composition or specifying the main components and their concentration is extremely important for reproducibility of the results.
Response: We thank the reviewer for this valuable comment. The description of the preparation and characterization of the components of the essential oils formulation was included at lines 111-129, as well as a descriptive table of the chromatography results at line 354.
Comment 2: Correct a typo in the caption to Figure 5: "48 and 72 hours before infection" → it should be "after infection".
Response: We thank the reviewer for this valuable comment. The text was corrected in line 349, Figure 5.
Comment 3: To clarify during the discussion whether the observed reduction in mortality (80% in T1 versus 100% in control) is sufficient for practical application in poultry farming.
Response: We thank the reviewer for this valuable comment. The text indicates that, despite the recorded mortality, a reduction in viral replication was observed during the experiment (lines 395-400). Additionally, the conclusions state that although this viral reduction is significant, in production conditions, efficiency requires better performance, so future experiments are needed to optimize these results in the field (lines 445-450).

Reviewer 2 Report
Comments and Suggestions for Authors
The manuscript entitled "Antiviral Activity of Essential Oils Against Avian Influenza Vi-2
rus H7N3 In Vitro And In Ovo Models" was well written. Furthermore, some points needs to take care for improvement:
In Introduction:
Give the research question clearly with the limitations of strategies in poultry and why EOs are used for the present study.
In methods:
Mention the rationale for the selection of the particular EOs. along with the oil authentication information.
Justify the stock solution concentration of 6000 ppm.
Justify the selection of conc range of 50 to 500 ppm for cytotoxicity study.
In discussion:
Give more discussion about the utilization of the EOs in the present study along with previously used as antiviral activity with references. It will be more better if author could mention the mechanism of action with the specific bioactive compounds.
Comments on the Quality of English Language
Very less typographical errors are there. Authors need to give proper attention.
Author Response
Comment 1: In Introduction: Give the research question clearly with the limitations of strategies in poultry and why EOs are used for the present study.
Response: We thank the reviewer for this valuable comment. A paragraph was added to this section to highlight the benefits of using essential oils in these tests. It covers their potential effects, the specific formulation proposed, and the rationale for further investigation to better understand their antiviral properties (lines 68-82).
Comment 2: In methods: Mention the rationale for the selection of the particular EOs, along with the oil authentication information. Justify the stock solution concentration of 6000 ppm. Justify the selection of conc range of 50 to 500 ppm for cytotoxicity study.
Response: We thank the reviewer for this valuable comment. The formulation and characterization methods for the essential oils used in this study are included, along with their previously reported antiviral effects, which were considered for their use (lines 111-130). It is noted that the dose used in this study was 4 times the supplier's recommended amount to assess the safety of this formulation in the embryonic model and its potential toxic effects (lines 132-134). Finally, lines 155-158 indicate that the dose used in the cytotoxicity test was determined based on the toxicity profiles of compounds with similar terpenes in other cell models.
Comment 3: In discussion: Give more discussion about the utilization of the EOs in the present study along with previously used as antiviral activity with references. It will be more better if author could mention the mechanism of action with the specific bioactive compounds.
Response: We thank the reviewer for this valuable comment. Information about the mechanism of action of essential oils was included, particularly those used during the present experiment, both in the introduction (lines 68-82), in the methodology (lines 111-114), and finally, a mention of the antiviral effect of essential oils, including avian influenza (lines 427-433).

Reviewer 3 Report
Comments and Suggestions for Authors
- The introduction doesn't explain “why” this specific combination of Lippia spp., Rosmarinus officinalis, and Eucalyptus globulus was chosen for testing against H7N3. While it mentions these plants have antimicrobial properties and lists their active compounds, there's no clear scientific justification for why this particular formulation would be expected to work against avian influenza.
- The introduction only briefly mentions two studies (Melissa officinalis against H9N2 and Fortunella margarita against H5N1) but doesn't adequately review what's already known about EOs against avian influenza viruses, making it difficult to understand the novelty of this work.
- The introduction fails to clearly articulate what specific knowledge gap this study addresses. It mentions that ongoing vaccination challenges exist and that new methodologies are needed, but doesn't explicitly state why existing EO studies are insufficient or what unique contribution this research makes.
- While the abstract mentions evaluating "a methodology suitable for assessing these compounds in more complex biological models," the introduction doesn't explain why the in ovo model is important, what advantages it offers over in vitro testing alone, or why this particular experimental approach is novel or necessary.
- The text mentions CPA was used for both in vitro and in ovo assays, while WT was only used for in vitro testing "as an internal control" - but what does "internal control" mean here? Were they comparing pathogenicity differences, or validating the method?
- The commercial product is described as a "topical spray nanoformulation," but readers cannot understand if 6000 ppm represents dilution from a more concentrated product or if this IS the product concentration.
- How were the "concentrations tested" (50, 100, 300, 500 ppm for cytotoxicity; then 3.12-100 ppm for antiviral) selected? The jump from cytotoxicity testing concentrations to much lower antiviral testing concentrations lacks explanation.
- Figure 4A shows mortality occurs at different timepoints (24h, 48h, 72h) with colored boxes indicating "HA positive at time of death" - but does this mean embryos that died earlier had higher viral loads? Were dead embryos immediately tested, or at fixed intervals?
- The text states "T1 showing the lowest levels (~10^11.4, p < 0.05)" and "No significant difference was observed between H7v (~10^11.61) and T2 (~10^11.8; p > 0.05)" - but p < 0.05 compared to what? What is the biological meaning of a ~1 log difference?
- The discussion repeatedly claims the EO formulation showed "antiviral activity" and "protective effect," but never clearly defines what threshold constitutes success when 100% mortality occurred in all groups by 72 hours.
- Multiple contradictory statements about treatment timing effects create confusion. First it says T1 "outperformed" T2, then says "EOs generally exhibit more substantial effects when used in early stages," then says T2 had "lower initial viral loads at 48 hours but more stability thereafter." Which treatment was actually better and why?
- The discussion mentions previous studies on Eucalyptus globulus, Rosmarinus officinalis, and Lippia spp., but never clearly states whether their IC50 values (20.4-38.3 ppm) are better, worse, or comparable to these studies.
- Looking at Figure 4A, T1 and T2 had 20% mortality at 24h while NTC2 had 10% - is 20% "minimal" or does it suggest early viral effect or EO toxicity?
- The discussion mentions validating "efficacy under field conditions" but never defines what this means for an in ovo study.
Author Response
Comment 1: The introduction doesn't explain “why” this specific combination of Lippia spp., Rosmarinus officinalis, and Eucalyptus globulus was chosen for testing against H7N3. While it mentions these plants have antimicrobial properties and lists their active compounds, there's no clear scientific justification for why this particular formulation would be expected to work against avian influenza.
Response: We thank the reviewer for this valuable comment. The Introduction has been revised to explicitly justify the selection of Lippia spp., Rosmarinus officinalis, and Eucalyptus globulus. These essential oils were selected due to their demonstrated complementary antiviral mechanisms against respiratory and enveloped viruses, such as influenza. Specifically, Lippia spp. interferes with viral adsorption and penetration; R. officinalis inhibits RNA virus replication; and E. globulus disrupts viral envelopes and promotes respiratory health. This information, along with a summary of prior research concerning essential oils active against avian influenza, has been incorporated into the revised manuscript (lines 68–82).
Comment 2: The introduction only briefly mentions two studies (Melissa officinalis against H9N2 and Fortunella margarita against H5N1) but doesn't adequately review what's already known about EOs against avian influenza viruses, making it difficult to understand the novelty of this work.
The introduction has been expanded to include a more comprehensive overview of previous studies investigating essential oils and their constituents in relation to avian influenza viruses. We now cite reports indicating that essential oils can inhibit viral membrane fusion, suppress NP and NS1 protein expression, and interfere with hemagglutinin–receptor binding. These findings collectively underscore the antiviral mechanisms relevant to avian influenza viruses. This expanded background provides greater clarity and positions our study as the first to assess a commercial essential oil formulation (comprising these three oils) against H7N3, using both in vitro and in ovo models. The revised text is presented in lines 74–82.
Comment 3: The introduction fails to clearly articulate what specific knowledge gap this study addresses. It mentions that ongoing vaccination challenges exist and that new methodologies are needed, but doesn't explicitly state why existing EO studies are insufficient or what unique contribution this research makes.
Response: We thank the reviewer for this valuable comment. The justification and knowledge deficiency addressed by this study are already articulated in the Introduction's concluding section (lines 83–98). This section elucidates that, although the implementation of existing vaccination and biosecurity measures, H7N3 remains a considerable concern for the chicken sector. It underscores the need for innovative, pragmatic antiviral strategies that are broad-spectrum, economically viable, and compatible with poultry management practices, including the use of essential oils. The paragraph specifies that the current study assesses a commercial essential oil formulation against AIV H7N3 utilizing both in vitro (MDCK) and in ovo (chicken embryo) models. We trust this part elucidates the intent and significance of our effort in accordance with the reviewer's recommendation.
Comment 4: While the abstract mentions evaluating "a methodology suitable for assessing these compounds in more complex biological models," the introduction doesn't explain why the in ovo model is important, what advantages it offers over in vitro testing alone, or why this particular experimental approach is novel or necessary.
Response: We thank the reviewer for this valuable comment. The introduction was modified in lines 100-106 to highlight the relevance of using chicken embryos due to their replicative capacity, making them a more robust and complex model than in vitro methods and allowing simultaneous or prior evaluation.
Comment 5: The text mentions CPA was used for both in vitro and in ovo assays, while WT was only used for in vitro testing "as an internal control" - but what does "internal control" mean here? Were they comparing pathogenicity differences, or validating the method?
Response: We thank the reviewer for this valuable comment. In the methodology section, lines 140-141, particularly in the Virus and Cells section, it is clarified that the WT virus is a biological reference and an internal control for comparing pathogenicity and replication with the CPA strain.
Comment 6: The commercial product is described as a "topical spray nanoformulation," but readers cannot determine whether 6000 ppm represents dilution from a more concentrated product or is the product concentration.
Response: We thank the reviewer for this valuable comment. In the section dedicated to the formulation and characterization of essential oils, the commercial reference of the product utilized in this experiment is provided, along with relevant patent information for additional technical consultation. Furthermore, its application specifies a concentration of 6000 ppm, which is 4 times the supplier's recommended level, to assess its safety and impact on chicken embryos (lines 132-134).
Comment 7: How were the "concentrations tested" (50, 100, 300, 500 ppm for cytotoxicity; then 3.12-100 ppm for antiviral) selected? The jump from cytotoxicity testing concentrations to much lower antiviral testing concentrations lacks explanation.
Response: We thank the reviewer for this valuable comment. A brief statement was included describing the levels of toxicity observed in the cytotoxicity test (lines 188-190), with the toxicity level at most 100 ppm, as previously evaluated.
Comment 8: Figure 4A shows mortality occurs at different timepoints (24h, 48h, 72h) with colored boxes indicating "HA positive at time of death" - but does this mean embryos that died earlier had higher viral loads? Were dead embryos immediately tested, or at fixed intervals?.
Response: We appreciate the reviewer’s insightful comment and agree that clarification was needed. We have now specified in the Figure 4 legend that each embryo was tested immediately after death was observed. Therefore, the label “HA positive at the time of death” indicates that hemagglutinating virus detection was performed on the allantoic fluid collected at the exact moment of embryo mortality, not at fixed time intervals. This clarification emphasizes that earlier deaths reflected a faster viral replication dynamic in the untreated control (H7v) compared with the essential oil–treated groups (T1 and T2), which showed delayed mortality and lower HA titers and viral loads over time (Section 3.3, Figure 4–5).
Comment 9: The text states "T1 showing the lowest levels (~10^11.4, p < 0.05)" and "No significant difference was observed between H7v (~10^11.61) and T2 (~10^11.8; p > 0.05)" - but p < 0.05 compared to what? What is the biological meaning of a ~1 log difference?.
Response: We thank the reviewer for their observation. The text was adjusted to specify that the p < 0.05 value corresponds to the comparison between the treated group (T2) and the untreated viral control (H7v) at 48 hours post-infection. Additionally, a clarification was included regarding the biological interpretation of the observed discrepancy. A decrease of nearly 1 log₁₀ in viral load (~91%) reflects a biologically significant reduction in viral replication, suggesting a nearly tenfold decrease in viral copy numbers, likely due to a lower viral replication rate in the embryo. The modifications were included in the updated text (Section 3.3, lines 297–304).
Comment 10: The discussion repeatedly claims the EO formulation showed "antiviral activity" and "protective effect," but never clearly defines what threshold constitutes success when 100% mortality occurred in all groups by 72 hours.
Response: Thank you for your comment and the opportunity to clarify this point. We understand that all embryos died after 72 hours. However, the revised discussion clearly states that the evidence of antiviral activity is based on the significant reduction in viral replication and the delay in early death observed in the treated groups.
The new text (Lines 390-408) indicates that the essential oil formulation reduced viral titers (EID₅₀, hemagglutination, and qPCR viral loads) and death rates 24–48 hours after infection compared to the viral control. These results suggest an initial antiviral effect on viral replication dynamics rather than a comprehensive protective effect on embryonic viability, consistent with the limitations of the in ovo model.
Additionally, the conclusion (lines 445-450) now highlights that the findings should be considered preliminary and require in vivo studies to assess safety, the appropriate dosage range, and efficacy in field conditions. The changes sufficiently address the reviewer's comments and clarify the biological criteria of success within the context of the model used.
Comment 11: Multiple contradictory statements about treatment timing effects create confusion. First it says T1 "outperformed" T2, then says "EOs generally exhibit more substantial effects when used in early stages," then says T2 had "lower initial viral loads at 48 hours but more stability thereafter." Which treatment was actually better and why?.
Response: Thank you for your comment. The revised version specifies that treatment T1 (pre-treatment with essential oils prior to viral infection) was the most effective antiviral modality, as it more substantially inhibited viral replication and reduced viral loads compared with T2 and the control (lines 390-408). Conversely, treatment T2 (virus pre-incubation with essential oils) also reduced viral replication, albeit to a lesser extent and more consistently, indicating a direct virucidal effect on viral particles. These observations are not contradictory; rather, they reflect different mechanisms of action contingent upon the timing of application: T1 functions prophylactically by obstructing the initial stages of infection (adsorption, penetration, and early replication), whereas T2 primarily exhibits virucidal activity (Lines 403-407). Overall, T1 demonstrated the highest efficacy, consistent with prior studies reporting the greater effectiveness of essential oils during the early stages of infection.
Comment 12: The discussion mentions previous studies on Eucalyptus globulus, Rosmarinus officinalis, and Lippia spp., but never clearly states whether their IC50 values (20.4-38.3 ppm) are better, worse, or comparable to these studies.
Response: We appreciate the comment. An explicit comparison of the IC50 values obtained in this study (20.4–38.3 ppm) with those previously reported for Eucalyptus globulus, Rosmarinus officinalis, and Lippia spp. (4–99 ppm) has been added to the revised version. The discussion indicates that the values observed here align with the ranges documented in the literature, as shown by the phrase: “supporting the comparable IC50 (20.4–38.3 ppm) range observed in this experiment." (Lines 423-426)
Therefore, our results confirm that the tested formulation demonstrates antiviral potency similar to that of the previously characterized essential oils, further supporting its potential as an antiviral agent against H7N3 AIV.
Comment 13: Looking at Figure 4A, T1 and T2 had 20% mortality at 24h while NTC2 had 10% - is 20% "minimal" or does it suggest early viral effect or EO toxicity?.
Response: We appreciate the reviewer’s observation. Although the T1 and T2 groups exhibited marginally higher early mortality rates (20% at 24 hours) than NTC2 (10%), this difference is not deemed biologically significant. It likely reflects early viral replication dynamics or transient stress associated with the inoculation procedure rather than EO toxicity. This interpretation is corroborated by the fact that embryo viability, morphology, and development remained normal throughout the study period, with no additional mortality observed at subsequent time points.
Comment 14: The discussion mentions validating "efficacy under field conditions" but never defines what this means for an in ovo study..
Response: We appreciate the reviewer's comment. Based on this, the conclusions section (lines 445-450) indicates the antiviral effect of essential oils in this model. However, the result is considered preliminary due to significant differences between the in ovo and in vivo models, and future studies in an in vivo model are required.
